# Drought Stress Study on *Nicotiana tabacum* L., "Baladi", an In Vitro Experimental Model

**Maria-Mihaela Antofie \*** and **Camelia Sava Sand**

Faculty of Agricultural Sciences, Food Engineering and Environment Protection, "Lucian Blaga" University of Sibiu, 550012 Sibiu, Romania; camelia.sava@ulbsibiu.ro
\* Correspondence: mihaela.antofie@ulbsibiu.ro

**Abstract:** Crops drought tolerance is a trait of outmost importance for agriculture especially today when climate change is affecting more the production for food and feed. The scope of this article is to evaluate in vitro drought stress response of *Nicotiana tabacum* L., "Baladi". The experiment was set up for four successive stages starting with in vitro seedling development, hypocotyl cultivation, three generations of micropropagation, pre-acclimatization and acclimatization. The effect of abscisic acid (ABA) and/or polyethylene-glycol 6000 (PEG) on tobacco hypocotyl caulogenesis and micropropagation were investigated. Superoxide-dismutases (SODs) and peroxidases (POXs) are more active and different isoforms patterns have been identified compared to the control for cualogenesis. A decrease of internodes length and a higher shoots multiplication rate were observed. However, under PEG treatment plantlets expressed hyperhydration and ceased rooting. Pre-treatments effects study of ABA and/or PEG were finalized in acclimatization phase for 18 tobacco clones. A summary of our results revealed that ABA and/or PEG induce among others a higher oxidative stress compared to the control in the first stage that is not maintained for all clones until acclimatization. Certain clones expressed a lower SOD activity compared to the control during acclimatization but maintaining higher POX activity.

**Keywords:** acclimatization; drought; hypocotyls; in vitro; peroxidase; shooting; superoxide-dismuthase; tobacco

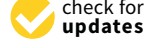



## 1. Introduction

At the global level we are facing the effects of climate change that for Europe have been predicted since 2010 [1]. According to a later study Romania, a European Union country placed into the Carpathian region is studied for dramatic droughts episodes for the future [2]. These scenarios were confirmed and drought episodes, more dramatic for Europe and beyond were consistently predicted since 2018 [3]. Biodiversity and agriculture are highly depending on natural factors and therefore it is considered that both of them are the most vulnerable towards climate change effects [4]. The term "biodiversity" includes all forms of genetic resources: wild, domesticated and new breeds. The most vulnerable crops resources need to be protected in situ and/or in gene banks for ensuring food security for the future [5]. Drought tolerance in plants will be one of the most challenging adaptation mechanisms, that usually acts as a combination of stress factors: biotic and abiotic [6,7]. Furthermore, it is highly recognized the concept of *stress matrix* developed by Mittler in 2006 [8] as acumulative effect of various abiotic and biotic stresses on plant growth and productivity. According to these authors the identification of physio-morphological traits that are affected by different combined stresses in crop plants that are the most important when breeding of these species is taken into consideration. The crops productivity remains relevant when considering strategies to avoid or face drought, other stress factors or a combination among them [9,10]. In the case of tobacco, that is an important economic crop [11], there are several scientific publications underlying the relevance of drought on

its production which is favoring the attack of pathogens too [12]. It is among the first scientific publications emphasizing a direct connectivity between abiotic and biotic factors for tobacco. Today we are witnessing the publication of solid and rigorous scientific literature related to the description of molecular mechanisms underlying drought tolerance and resistance in plants [13]. Thus, abscisic acid (ABA) and ethylene are considered as the main molecules highly involved in controlling drought tolerance in plants [14]. Additionally, considered of high value are, also, osmoprotectors, plant antioxidative capacity as well as desiccation tolerance [15]. However, the major challenge for breeders remains plant selection [16,17]. The selection of in vitro obtained germoplasm continue to receive attention even the genetic modification in plants gained today a huge interest mostly related to clustered regularly interspaced short palindromic repeats (CRISPR) technology [18,19]. In vitro multiplication continues to be an essential part for the entire technological flow in plants breeding [20–22]. On the other hand, the continuous in vitro plant's experimentation may reveal new insights related to the connectivity between drought tolerance and morphogenesis. In this regard, ABA signaling pathway raise questions related to other plant development processes interference than stomatal function or seed dormancy, already extensively studied [23]. Thus, complex plant morphogenetic processes such as caulogenesis or shoot formation, rooting, decreasing internodal length still need to be studied to understand the genetic plasticity of plants towards drought tolerance [24,25]. On contrary polyethylene-glycol or PEG was used for inducing drought stress in plants since the late 1970s [26]. Molecular mechanisms of drought signal transduction due to PEG based on osmotic pressure exerted on plasma membrane is already described [27]. Moreover, the ABA role of downregulating gene expression for PEG osmotic stress signaling pathway was depicted [28]. The complex mechanisms of reactive oxygen species burst (ROS) related to drought induced by ABA or PEG was also described [29]. It is well established that ROS triggers the activation of antioxidant enzymes activities such as superoxide-dismutase (SOD) catalase (CAT), peroxidase (POX) and glutathione-reductase (GR), making relevant the analysis of these enzymes for drought stress [30].

Based on these evidences, the scope of this article is to study the effect of both: abscisic acid (ABA) and polyethylene-glycol 6000 (PEG) on tobacco (*Nicotiana tabacum* L., "Baladi") by in vitro cultivation for long term. The study starts with in vitro shoots regeneration from hypocotyls up to acclimatization, aiming the analysis of two major categories of enzymes: superoxide-dismutase (SODs) and peroxidase (POXs). Caulogenesis, multiplication rate and internodes length were analyzed too for ABA and/or PEG effects and will be discussed in correlation with the activation status of SODs and POXs.

## 2. Materials and Methods

### 2.1. Plant Material

Tobacco seeds, *Nicotiana tabacum* L., "Baladi", of commercial origin were used. They were disinfected into the laminar flow hood by using 70% ethanol for 3 min followed by immersion into a 1.2% sodium hypochlorite solution supplemented with 0.1% Triton X-100 for 10 min. In the end three rinses with sterile water for 1 min each were realized before sowing for germination into 6 cm in diameter Petri dishes filled with 12 mL Murashige-Skoog solidified culture medium (MS 1962) without hormones [31] (stage zero, Table 1). After seedlings formation, explants of tobacco's hypocotyl, of about 8 mm length, were harvested and inoculated on four variants of modified culture media MS 62 (stage two, Table 1; 4 hypocotyl per Petri dish and 5 Petry dishes per variant with a total of 20 hypocotyls per variant). Plantlets as shoots originating from hypocotyls were further sub-cultivated for three generations on experimental modified culture media MS 62 stage 2. Only shoots over 1 cm have been harvested for cultivation and their origin was monitored. In the end 20 shoots per variant were harvested and cultivated. Each variant followed the same culture media characteristics such as control and MS 62 supplemented with ABA and/or PEG6000 (stage three, Table 1). After the first shoot cultivation a total of 18 clones of tobacco plantlets were isolated, by taking into consideration their origin, and further

monitored starting with the stage two as following: 2 ABA, 11 PEG, 4 ABA+PEG and 1 for control. The uniformity of regenerated shoots in terms of height per test tube where among the major characteristics for clone selection (i.e., over 1 cm). Based on our experience it was considered that control cannot influence the new shoot formation for drought and, as a consequence, the selection was for a single control randomized selected test tube. For pre-acclimatization all plantlets were sub-cultivated on MS 62 without hormones (stage 3, see Table 1).

**Table 1.** Cultivation conditions of *Nicotiana tabacum* L., "Baladi" to investigate ABA and PEG effects in four experimental stages. The classic Murashige-Skoog culture medium was or not supplemented in different conditions with benzyl-aminopurine (BAP), naphthalene-acetic acid (NAA), abscisic acid (ABA) and polyethylene-glycol 6000 (PEG).

| Stages | Explant | MS 62 without Hormones | MS 62 + BAP (1 mg/L) NAA (0.1 mg/L) | ABA (20 μM) | PEG6000 50 g/L | ABA 20 μM PEG6000 50 g/L |
|---|---|---|---|---|---|---|
| Stage 0 | Seeds | ✓ | | | | |
| Stage 1 | Hypocotyls | | ✓ | ✓ | ✓ | ✓ |
| Stage 2 | Shoots multiplication for three generations | ✓ | | ✓ | ✓ | ✓ |
| Stage 3 | Shoots for rooting in pre-acclimatization | ✓ | | | | |
| Stage 4 | Plantlets for acclimatization | ✓ | | | | |

### 2.2. In Vitro Culture Media

Basic culture media used in this experiment were based on Murashige-Skoog 1962 or MS 62 and different variants have been used for different stages.

For seed germination a MS 62 culture medium without hormones was used.

In the case of stage 1 of the experiment, modified MS 62 culture media were used, supplemented with 6-Benzylaminopurine or BAP (1 mg/L) and 1-Naphthaleneacetic acid or ANA (0.1 mg/L). Four variants were used: control, modified MS 62 medium supplemented with abscisic acid or ABA (20 μM) and/or polyethylene glycol or PEG6000 (50 g/L). All reagents used in culture media preparation were of Sigma-Aldrich origin.

In the case of stage 2 of the experiment, four types of MS 62 culture media without hormones and supplemented with ABA (20 μM) and/or PEG6000 (50 g/L) were used for micro-propagating tobacco plantlets for three generations. Pre-acclimatization consisted in the plantlets' cultivation on MS 62 basal medium without hormones to ensure rooting (stage 3, see Table 1).

### 2.3. Basal Conditions and Materials for In Vitro Culture and Acclimatization

For stages 0 and 1, Petri dishes of 6 cm diameter were used filled in with 12 mL culture medium each. For stage 2, test tubes of 20 mL filled in with 5 mL of culture medium were used. Overall, 20 replicates/variant were used for each variant and stage.

Inside the cultivation rooms the major standard parameters are as following: temperature 22–24 °C, a photoperiod of 16/8 h; light/dark, a photon flux density of 53.8 μmol/m²s and a humidity in the air of 30%.

As a substate in acclimatization (the fourth stage, see Table 1) was used perlite and a sterile MS salt basal nutritive solution, disposed in a layer of 5 cm into squared trays of 30 cm on each side. It was ensured a temperature of 22–24 °C, a photoperiod of 16/8 h; light/dark and a photon flux density of 53.8 μmol/m²s.

### 2.4. Biochemical Analysis

Biochemical analysis consisted in the spectrophotometric assessment of protein content according to Bradford [32], superoxide-dismutase (SOD) activity according to Giannopolitis and Ries [33] (a unit of enzyme activity is defined as the amount of enzyme required to inhibit the Nitrotetrazolium blue chloride reduction reaction by 50%, following reading at 560 nm) and peroxidase activity POX that was recorded according to Baaziz and Saaidi [34] at $\Delta$ A 470 nm/min mL. For each variant a three times repetitions were realized through randomized selection of at least three samples, and statistic methods were applied for results interpretation. All reagents are of Sigma-Aldrich origin.

Tobacco hypocotyls explants of 1 g were ground using a mortar pestle in phosphate buffer of 0.1 M at pH 7 and 4 °C (3:1 $w/v$), centrifugated 20 min. at 18,000 rpm. The filtrates were precipitated by adding 90 % acetone ($v/v$) under continuous stirring for 2 h. The obtained solution was centrifugated at 15,000 rpm for 15 min and pellets were solubilized in Tris-HCl buffer pH 6.9 50 mM. The mixture was cooled overnight at +2 °C for SODs and POXs analysis.

Vertical electrophoresis parameters for POXs: +4 °C, 20 mA, 3 h. Gel for electrophoresis: 8% of polyacrylamide (PAA) and the stacking gel 4% PAA and buffer Tris-Gly 0.05 M (pH 8.3) with a Sigma vertical electrophoresis unit. Bromphenol blue was used as a running maker. POX activity was revealed by incubating gels into a solutions of 0.5 M acetate buffer at a pH 5 supplemented with 0.08% benzidine and couples of drops of 30% $H_2O_2$.

Vertical electrophoresis parameters for SODs: +4 °C, 20 mA, 60 min. Gel for electrophoresis: 10% of polyacrylamide (PAA) and the stacking gel 5% PAA and buffer Tris-Gly 0.05 M (pH 8.3) with a Sigma vertical electrophoresis unit. Steps in revealing SODs: immersion in Nitroblue tetrazolium (NTB) 1.23 mM for 15 min, rinsing, immersion into potassium phosphate buffer, pH 7.0, supplemented with 28 mM tetramethylenediamine (TEMED) and $2.8 \times 10^{-2}$ mM riboflavin for 15 min in dark conditions, rinsing and illumination under 30 mE/m²s for 15 min. All procedures were carried out at room temperature and all materials were previously refrigerated at +4 °C and following the method described before [35].

### 2.5. Morphometry and Statistical Data Analysis

All morphometric measurements and statistical analyses were realized on at least 20 repetitions/variant and consisted in determining: fresh weight, number of shoots (i.e., over 10 mm in heigh) and the internodal distance as the ratio between the plant height and the number of nodes expressed in cm by weight classes of regenerants. All measurements were repeated three times for different individual plants for biochemical analysis. Analysis by one way ANOVA test approach was performed using Excel for Windows by the least significant difference (LSD) test at 0.05 probability level when the F test showed a significant ($p \leq 0.5$) effect.

## 3. Results

### 3.1. ABA and/or PEG Effect on Caulogenesis

#### 3.1.1. First Stage and Control

Tobacco seedlings were easily obtained after 3 weeks of in vitro seeds cultivation based on the culture media recommended by Murashige Skoog (1962). No infection was recorded. During 3 weeks of cultivation seedlings developed well for almost 2 cm length and they were prepared for further cultivation on MS 62, stage 1, of the experiment according to Table 1. Seedlings' hypocotyls of around 8 mm length were cut-off at the limits of roots and cotyledons and subcultivated on Petri dishes for four variants of MS 62 culture media supplemented with ABA and/or PEG (Table 1 stage 2) following the protocol described in 1990 by Ebert and Clarke [36]. At this stage it was important to add a cytokinin (BAP) and an auxin (NAA) in order to support dedifferentiation and cell differentiation for shoots organogenesis [37]. Following this stage cytokinin and auxin were no longer needed according to Murashige and Skoog studies for micropropagation [31].

Five classes of fresh weighs caulogenetic hypocotyls were obtained in the end of four weeks of in vitro cultivation ranging between 0.5 and 4.8 g and a total rate of 48.67 shoots per hypocotyl for control (Figure 1a). SODs and POXs activities and expressions are lowest compared to the other variants (Figures 2 and 3). SOD activity was 4.50 UAE and POX activity 10.61 μmol $H_2O_2$ (Δ 470 nm/mL min).

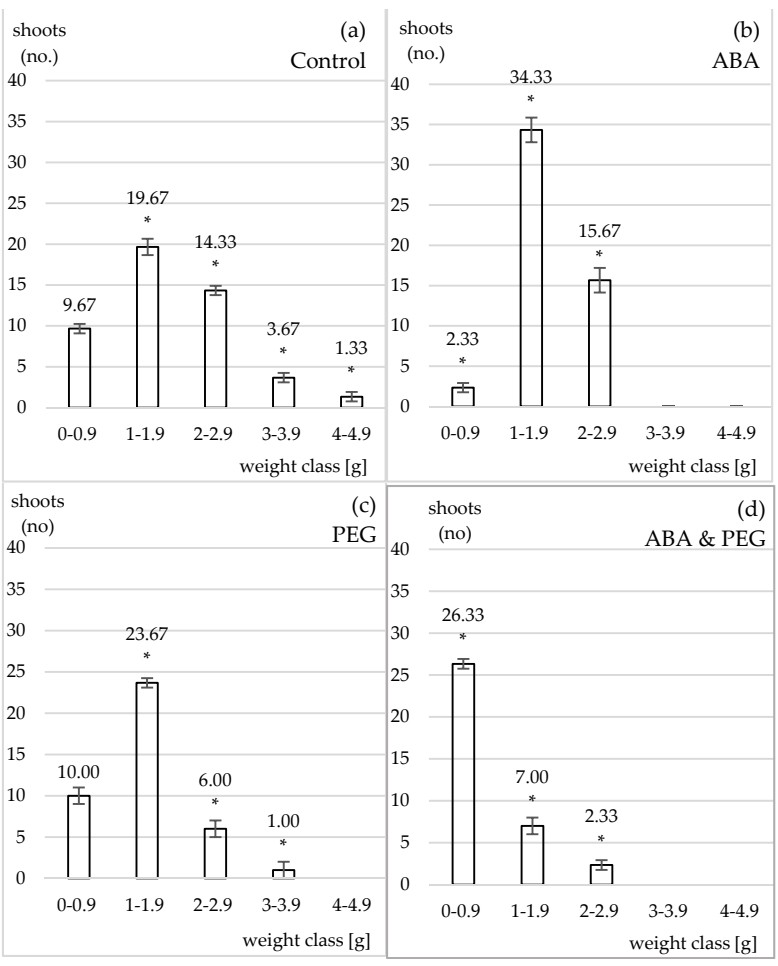

**Figure 1.** The shooting rate on fresh weight classes of morphogenetic hypocotyl of *Nicotiana tabacum* L., 'Baladi' was ceased for the highest class of fresh weight in case of control (**a**) compared to all variants cultivated on MS 62 modified culture media supplemented with hormones and ABA (**b**), PEG (**c**) and ABA & PEG (**d**) (stage no 1). * Significant differences: $p < 0.05$.

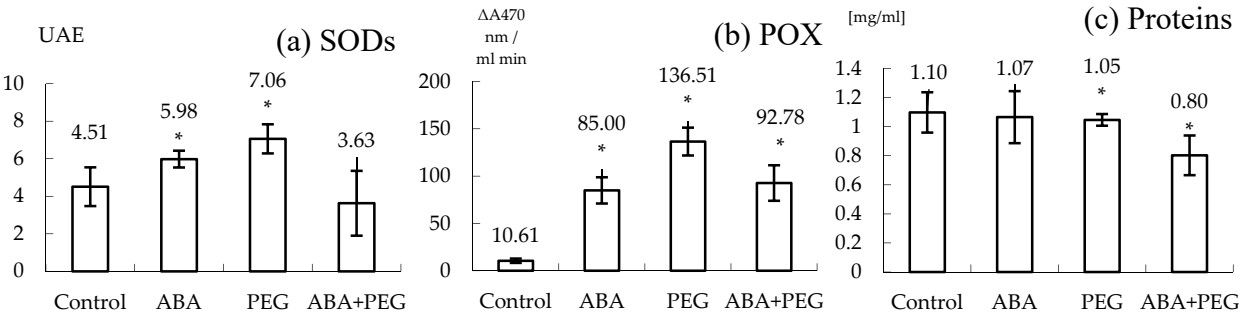

**Figure 2.** The effect of ABA and/or PEG on the SODs and POXs activities and on total protein content of the shooting hypocotyls in tobacco (*N. tabacum* L., 'Baladi'). A significant increase was recorded for SODs activity when ABA and PEG were added into the culture medium (**a**); A significant increase of POX activity was recorded for ABA and/or PEG (**b**). Proteins were significantly decreased only under both stressor presence: ABA and PEG (**c**). * Significant differences: <0.05.

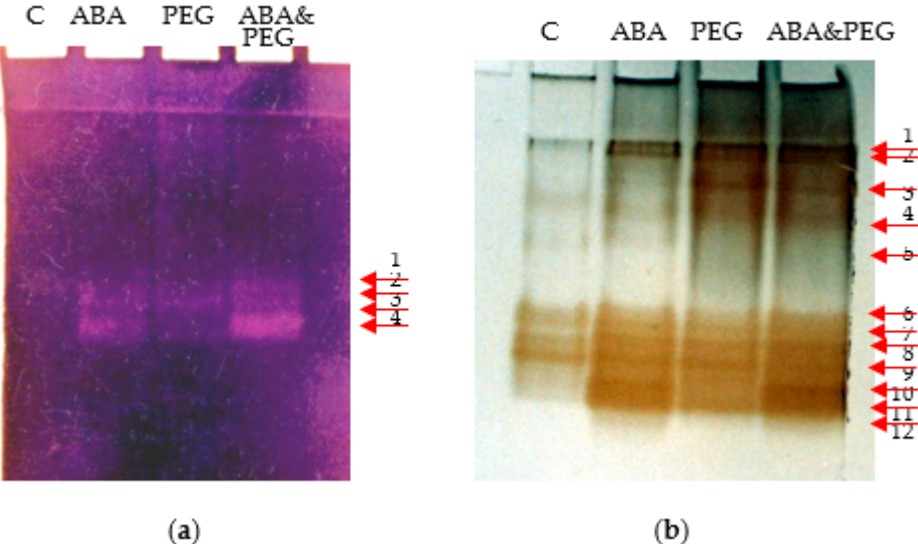

**Figure 3.** SODs (**a**) and POXs (**b**) isoforms revealed into electrophoretic gels of tobacco hypocotyls (*N. tabacum* L., "Baladi") that have been submitted to the effect of ABA, PEG and ABA & PEG. In the right there are red arrows to emphasize the expression of isoforms after electrophoresis. C: control.

### 3.1.2. ABA Effects on Tobacco Caulogenesis

Culture medium supplemented with ABA induced a fast effect on increasing the number of shoots for the second class of weight (34.33 shoots per hypocotyl) as well as per total compared to the control and the other variants (52.33 shoots per hypocotyl). It was obviously the lack of the last two heaviest weighing classes (Figure 1b). ABA induced a strong and significant response for SODs and POXs activity as well as for the SODs and POXs isoforms observed upon electrophoresis (Figures 2 and 3).

### 3.1.3. PEG Effects on Tobacco Caulogenesis

In this experiment, 5% (*w/v*) PEG 6000 would induce an osmotic potential of approximately −0.4988 (according to the formula developed in 1973 by Michel and Kaufmann, from the Department of Botany of the University of Georgia, USA) in distilled water, and in culture media supplemented with agar it would not exceed the average value of the water deficit [38]. Only PEG alone was able to induce the generation of the first fourth classes, that is closely connected to the drought stress induced for tissues at the contact with culture media and a total media of 40.67 shoots per hypocotyl (Figure 1c). In addition, the expression of hyperhydration is general, as well as the absence of roots. It was recorded an increase in the activity and expression of SODs and POXs compared to control, given that the protein concentration did not significantly change (Figures 2 and 3).

### 3.1.4. ABA & PEG Effects on Tobacco Caulogenesis

Culture medium supplemented with ABA & PEG apparently induced a higher drought effect, as only first three weigh classes of caulogenetic hypocotyls were obtained. ABA & PEG induced a total rate of 35.66 shoots per hypocotyl (i.e., bellow 3.9 g). By associating ABA as the natural mediator of drought stress with PEG, the external osmotic stress inducer, the negative effect manifested on the size of explants as well as caulogenesis is proved. In this case, the average production of shoots is 9.27/explant or 11.9/g of explant, the present weight classes ranging between 0 and 2 g, the others disappearing. Two characteristics have been noted: hyperhidrosis on transformed hypocotyls and roots absence for the new formed shoots. From a biochemical point of view, these effects are associated with an increase in POXs activities but apparently not in SODs, although the analysis of electrophoretic spectrum reveals certain emphasizing of SODs' isoforms expression (Figures 2 and 3).

### 3.2. ABA and/or PEG Effect on Multiplication

In total, 18 tobacco clones, generated from the same second class's weight of hypocotyls, were selected for continuing the experiment for micro-propagation (stage 2, see Table 1). The main characteristic was the class of weight of hypocotyls as well as a single collected shoot per test tube. After the first sub-cultivation it was settled the following clones: 2 for ABA, 11 for PEG, 4 for ABA & PEG and 1 for control. In the end of stage 2 the highest multiplication rate was obtained for the shoots originating from hypocotyls previously treated both with ABA & PEG. This rate is higher compared to control, also compared to ABA or PEG. Moreover, it is obvious the decrease of the height of plantlets and this is due to the internode length decrease (Figure 4).

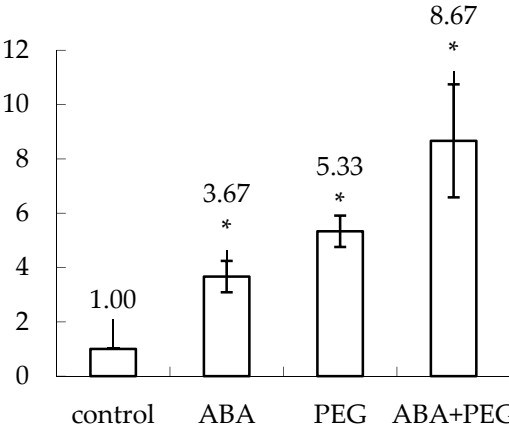

**Figure 4.** The rate of micropropagation of tobacco shoots *(N. tabacum* L., 'Baladi') clones originating from hypocotyls submitted to the effect of ABA (clones 1 and 2), PEG (clones 3–13), ABA & PEG (clones 14–17) and control (C) were monitored during three generation of cultivation in stage 2. It is obvious a stimulation of the multiplication rate of all plantlets that have been generated from stressed hypocotyls with ABA, PEG and ABA & PEG. * Significant differences: $p < 0.05$.

### 3.3. ABA and/or PEG Pre-Treatment Effects on Acclimatization

Acclimatization was part of the stage 4 of our experiment when plants are re-adapting towards natural environmental conditions (see Table 1). This process is usually carried out gradually, for 3–4 weeks, the main technical parameter being the gradual reduction of atmospheric humidity from 100% (i.e., in vitro cultivation conditions) down to 40% (average greenhouse humidity) and down to 30–20% (natural conditions of humidity in the temperate-continental environment of Romania). The most critical period during the acclimatization phase is between the zero moment and the 10th day, when it is considered that the plants completely consume their energetic reserves during the pre-acclimatization— in vitro. The technical parameters used to control this stage are: humidity, temperature, pH of the substrate solution used, substrate characteristics, light intensity and photoperiod. As no plant losses were recorded during acclimatization, it is considered that this stage was performed in optimal conditions. Biochemical and morphometric analyses were performed at 10 days after the start of acclimatization. It was not realized enzymes electrophoresis due to the complexity of work (i.e., 18 types of samples). In this regard, the length of internodes was evaluated for acclimatized plants (Figure 5) as well as the activity of SODs and POXs and protein content. In this stage, 18 clones represented by 20 plants of each have been investigated and the previous presence of ABA, PEG and ABA & PEG during acclimatization was investigated.

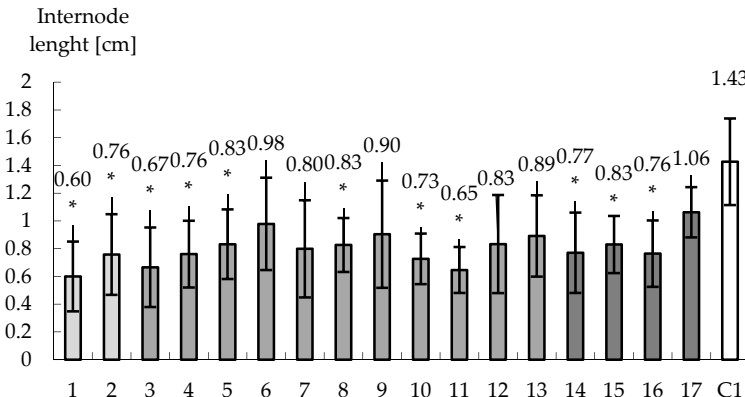

**Figure 5.** The evolution of internodes length for 18 tobacco clones studied and originating from hypocotyls that have been submitted to drought stress such as ABA (1 and 2), PEG (3–13), ABA & PEG (14–17). It is obvious the significant decrease in internodes length and in certain clones at half compared to control. * Significant differences: $p < 0.05$.

## 4. Discussion

The effect of drought on plant morphogenesis is difficult to be studied because of the oxidative stress matrix acting in any morphogenetic processes, and there can be an overlapping effect between drought and morphogenesis. However, in vitro culture may provide us new insight regarding the impact of ABA and PEG on morphogenesis [39]. The study of the complex enzymes system acting (i.e., superoxide-dismutase (SOD) catalase (CAT), peroxidase (POX), glutathione-reductase (GR), is well recognized to control the ROS generated levels during oxidative stress [30]. SODs and POXs were investigated for their activity related to the effect of ABA and/or PEG6000 (PEG) on caulogenesis, micropropagation as well as for acclimatized plantlets of tobacco. The complexity of this experiment is that the oxidative stress recorded for both types of studied enzymes is coupling morphogenesis (i.e., caulogenesis, rooting and shoot multiplication) with the activation of drought stress.

### 4.1. ABA and/or PEG Effects on Caulogenesis

The general idea of the experimentation was to use as explants tobacco hypocotyls to study the potential of shoot formation or caulogenesis under drought factors such as ABA and PEG. Thus, it was studied the oxidative stress that is associated to morphogenesis on one hand and drought stress on the other hand for a long period of time. For tobacco, hypocotyls are considered as moderate morphogenetic tissues compared to leaf fragments but generally are well known for their morphogenetic response expressed into in vitro conditions [39]. As a consequence, the oxidative stress that may be recorded for morphogenetic processes development may be generated by the presence into the culture medium of ABA on one hand and by PEG on the other hand. ABA is well known as a hormone that is also supporting morphogenesis process as well as the answer of plants for different stress factors [23]. Thus, ABA at a concentration of 20 μM was used in previous studies on other tobacco explants and it was considered as appropriate to be used in this experiment both for supporting morphogenesis as well as for activating drought stress [23,40,41]. In the case of PEG6000, we used the same concentration of 5% (*w/v*) applied by other authors and considered as moderate in terms of osmotic stress [42]. Thus, the major effect of this substance is to activate the osmotic stress. We may add that soil water loss is thought to induce a decrease in osmotic potential from 0 to −1.5 MPa when water loss is total, in order to ground the concentration of PEG6000 used in this experiment [43]. Most areas in Europe and generally in temperate regions area considered as "middle soils" related to water deficit, i.e., −0.4 and −0.6 MPa. Thus, this low concentration was tested due to the longer duration of explants cultivation on one hand and due to the scope of this experiment to study tobacco caulogenesis under the effect of ABA and/or PEG. After four weeks of

in vitro cultivation, all hypocotyls were able to produce shoots. This result may ground the idea that the chosen concentrations for all used substances worked at the physiological level for in vitro cultivation of tobacco hypocotyls on one hand and that they are activating the morphogenetic potential of this tobacco tissue on the other hand. However, rooting was absent when PEG was added into the culture media. Rooting was recorded for all ABA treated plantlets. This may be related to the osmotic potential activation that is negatively influence rooting, which usually take place inside the culture medium.

As a general remark control hypocotyls generated five classes of caulogenetic hypocotyls weighing between 0.5 and 4.8 g. As the number of developed shoots on each of explant significantly varied it was considered that it will be relevant to assess the shoot number per each class of weight. The first observation was that all the other experimented culture media (i.e., supplemented with ABA and/or PEG) do not generated the last weighing class and this may be connected to the activation of drought stress that was detrimental to morphogenetic processes to a certain extent (i.e., over 4 g). Thus, hypocotyls clearly developed shoots, their morphogenetic potential was not completely inhibited or arrested by supplementing culture media with ABA and/or PEG. At this stage it was needed to add cytokinin and auxin and this is in line with other authors because the morphogenetic potential implies complex processes such as dedifferentiation and redifferentiation as well as cell division and enlargement processes that are supported by a right balance between cytokinin and auxin [31,37].

In the case of ABA, the analysis of results presented in Figure 1 reveals that this hormone induced the most significant morphogenetic response as for all replicates the weight were no heavier than 2.9 g. It can be considered the loss of water to be associate to the activation of drought stress. However, the lack of other supplementary external factors, do not impede hypocotyls to undergo shooting and rooting. The analysis of these results revealed that ABA support the increase of shoot formation compared to the control group in tobacco hypocotyls that is in line with other studies [44]. Thus, Wang and collaborators also discuss about the loss in weight and in the height of plantlets. In another case it was studied the phenomenon on indirect organogenesis of *Zea mays* and considered that ABA, exogenously administered at concentrations between 10–20 μM, can induce positive effects on the regeneration process, and seems that this is also the case of tobacco [45]. Additionally, along with the hypocotyls, practically all explants showed rooting processes, their general appearance being comparable to that of control. From biochemical point of view ABA induced a significant increase for SODs and POXs compared to the control (Figure 2a,b). In this case, it can be considered that ABA clearly activated oxidative enzymes more compared to control. The drought stress clearly induced the producing of ROS as the SODs are 1.3 higher in their activity compared to control. Moreover, the POXs looks to become more active too becoming 1.5 more active compared to control. Total protein content is similar to the control in case of ABA treated hypocotyls. The analysis of electrophoretic spectrum reveals that SODs isoforms appear more active compared to the control too and, isoforms 3 and 4 are more concentrated compared to the control (Figure 3a). In the case of POXs electrophoretic spectrum analysis, it is relevant to underline the new expression of lighter isoforms 9–12 and the middle weight such as isoform 5 (Figure 3b). As caulogenesis and rooting are not blocked it may be considered that these biochemical answers are supporting morphogenetic potential of hypocotyls and may be relevant for the in vitro adaptation to drought stress induced by ABA. In this particular case, we may consider that the oxidative stress activated for drought by ABA can be supported by the new iso-forms pattern expression for POXs and SODs, but the missing of acting another drought stress factor may explain why the content of protein is similar to control.

Direct caulogenesis is negatively influenced by PEG6000 5% (*w/v*), inducing a shoot rate of 10.49/g explant, or 14.36/explant (Figure 1c) and a reduced size of shoots in accordance with other research results [43]. However, it is relevant to emphasize, the general manifestation of hyperhydration, as well as the absence of roots for all generated shoots. The manifestation of the hyperhydration is positively correlated with the decrease

of fresh weight, a character that has been emphasized by other authors who have worked on other in vivo experimental models [46,47]. In this case also four classes of caulogenetic hypocotyls have been obtained, the heaviest being absent (Figure 1c). Hyperhydration contributed to water logging in the explants if we consider the total protein content that is significantly lower for all explants (Figure 2c). In this case also, PEG induces the highest multiplication rate for the second class of weight (Figure 1c). It was recorded that there was an increase in the activity and expression of SODs and POXs compared to the control, given that the protein concentration does not significantly change. Thus, SOD activity was higher compared to the control and ABA treated hypocotyls (Figure 2a). It is obvious the slight expression of SODs isoforms 3 and 4 compared to the control but lighter compared to ABA. On contrary POX activity was more important: 13 times higher compared to the control and more than 1.6 higher compared to ABA effect (Figure 2b). Considering the POX isoforms, it appears that the heavier are more well expressed (i.e., isoforms 1–4) compared to the control. The stronger effect of PEG presence appears to be shared at least between SOD and POX enzymes. However, it appears that POX activity is significantly stronger, and it may be associated with the hydric effect in inducing hyperhydration an effect already described by other authors [48]. PEG appears to increase caulogenesis compared to the control and is restraining the weight range of caulogenetic hypocotyls by excluding the heaviest class. However, it appears also that POXs are more activated for inducing the drought stress and apparently there should be shared between water logging and water loss considering hyperhydration and shooting. The increase of osmotic potential may be detrimental for rooting and, also, may be responsible for this higher oxidative stress.

Caulogenesis appears not to be blocked under the effect both of ABA & PEG. However, the highest rate of shoot formation was recorded for the lighter class of cualogenetic hypocotyls and the last, the heavier classes being not recorded (Figure 1d). In this case browning of caulogenetic hypocotyls were observed besides the constant presence of hyperhydration and lack of roots. The analysis of SOD activity reveals a decrease in its activity compared to the control (Figure 2a) associated with a decrease in total proteins. The latest may be corelated to the hyperhydration. However, the expression of SOD isoforms 3 and 4, is obvious and almost similar to those obtained for ABA effect. POX activity is significantly higher compared to the control, but it is comparable to ABA effect. In this case a stronger answer is given to almost all POX isoforms: medium as well as lighter isoforms being very well expressed compared to all other variants. Considering a significant reduction in protein synthesis, then this result could be due to hyperhydration and being in line with other studies [49]. Finally PEG & ABA, exacerbates the effect of drought stress at hypocotyl level compared to culture media supplemented either with ABA or PEG.

Summarizing it can be considered that the oxidative stress burst generated in the presence of ABA is moderate compared to that of PEG as in this case hypocotyls were able to generate the highest number of shoots and induced rooting. PEG is an osmotic stressor and is inducing a higher impact on the onset of oxidative stress. In this regard the hypocotyl explants are able further to develop shoots but at a moderate level compared to those treated with ABA. Moreover, hyperhidrosis and the lack of roots were observed proving that the oxidative stress activated by PEG is arresting rooting. In this regard, the protein content seems to be decreased only for culture medium supplemented with ABA & PEG and this biochemical answer may be the effect of adding to ABA another extra osmotic stress. By studying the electrophoretic spectra of both enzymes, it was revealed, in case of SODs that four major bands are well expressed for stressed variants compared to control. The close qualitative analysis of these results revealed that the strongest answer is associated to ABA as well as to ABA & PEG. It is obvious that at least bands 3 and 4 of SODs are well expressed for the drought stressed hypocotyls compared to control. In the case of POXs, 12 electrophoretic bands have been revealed and only 7 are characterizing the control (i.e., the isoforms: 1, 2, 4, 5, 6, 7, 8, 9). In the case of stressed hypocotyls, other new 5 isoforms are well expressed especially of low weight that are isoforms from 6 to 12.

Based on these results analysis it appears that POXs isoforms 11 and 12 for tobacco are highly supported in their expression by ABA. At moderate level they are also expressed due to osmotic stressed induced by PEG. However, in case of both stressors, it seems that the expression of all isoforms is more uniform and stronger even the total measured activity is lower compared to the effect of PEG but comparable to the effect of ABA. Such a inconsistent biochemical answer related to the study of oxidative enzymes can support the restraining of the range of weighing classes for stressed hypocotyls on one hand and the best shooting rate in case of ABA. It can be considered that in this particular case, both ABA and PEG, act synergically when their concentrations are well balanced from physiological point of view. Furthermore, these biochemical analyses will support us to understand that hyperhydration as well as the lack of roots are direct answers to a higher oxidative stress produced by PEG as well as by ABA & PEG.

*4.2. ABA and/or PEG Effects on In Vitro Multiplication*

A suite of 18 clones have been monitored for three generations of multiplication considering the same culture media supplemented with ABA and/or PEG but hormone free. The reason to multiply without NAA and BAP supplemented into the culture medium is that tobacco shoots need no hormones to grow [31]. Moreover, we were interested in removing as much as possible substances that may further influence our analysis compared to the studied literature on tobacco. Control plantlets developed up to 8 leaves and a single shoot per test tube that is in line with other studies [31]. In the case of ABA and/or PEG supplemented media, the most common phenomenon is the organogenetic callus appearance at the level of detachment of shoots originating from hypocotyls that are continuously treated with the same drought inducer for three generation, with the exception of control plantlets. It may be associated with the wounding stress that become more exacerbate when osmotic and drought stress factors are supplemented in the culture media as it was noticed before [50]. The multiplication rate was analyzed based on the number of new generated shoots and therefore the control is 1 (one) connected to the single shoot generated (Figure 4) as the plantlets developed in height by generating new nodes and leaves. The best multiplication rate was recorded for the variant cultivated on ABA & PEG. The internodal distance is significantly lower in all experimental clones compared to control. This parameter becomes important for drought-resistant plants, as the reduction of the plant size correlates positively with the reduction of the transpiration surface. In this particular case, we may consider that the oxidative stress worked also for supporting morphogenesis implying and controlling dedifferentiation, redifferentiation, cell division and elongation. The pre-acclimatization phase was set in the absence of any stressor by cultivating plantlets only on MS 62 without hormones. Each explant generated a single complete plantlet of tobacco ready to be moved in acclimatization as they developed roots and were able to be transferred into acclimatization at the end of four weeks of cultivation. Thus, ABA and/or PEG pre-treatment do not have any further effects such as: arresting rooting, plantlets are not presenting hyperhydration or necrosis any longer.

*4.3. ABA and/or PEG Pre-Treatment Effects on Acclimatization*

The acclimatization phase is highly challenging for the complete success of plant biotechnology, before plants should be transferred to greenhouse [51]. Drought stress is among the major external factors that influence acclimatization [52–55]. In this study all 18 tobacco clones have been investigated for their ability to face the stress of acclimatization. This process is named today drought memory and it is in line with other recent studies (osmotic stress) [56], for ABA effect [57]. Scientific results and hypothesis are already supporting this concept [58]. All plantlets were able to pass acclimatization. However, the heigh of all pre-treated plantlets with ABA and/or PEG showed significant decrease based on measurements of the first three internodes of each plant and 20 plants per variant. Thus, significantly dwarf plantlets were obtained for both ABA clones (i.e., 0.6 and 0.76 cm). However, in this case only for the first clone the SOD activity was significantly lower

compared to control. The second clone expressed a higher SOD activity compared to the control (Figure 6a). POXs activity and total protein contents follows the same pattern compared to the control (Figure 6b,c). It may be considered that the first clone seems to face easier the transfer to acclimatization considering the SODs activity. However, the activation of oxidative stress induced by ABA works at different levels. If into in vitro conditions stomata are not working in acclimatation they are compulsory to function and ABA is controlling their functioning. It is possible that such a different answer between the two clones to modify the expression of ABA downregulated genes for stomata function, but our results cannot substantiate this. If such a hypothesis should be that plant drought tolerance need further to be oriented in this regard.

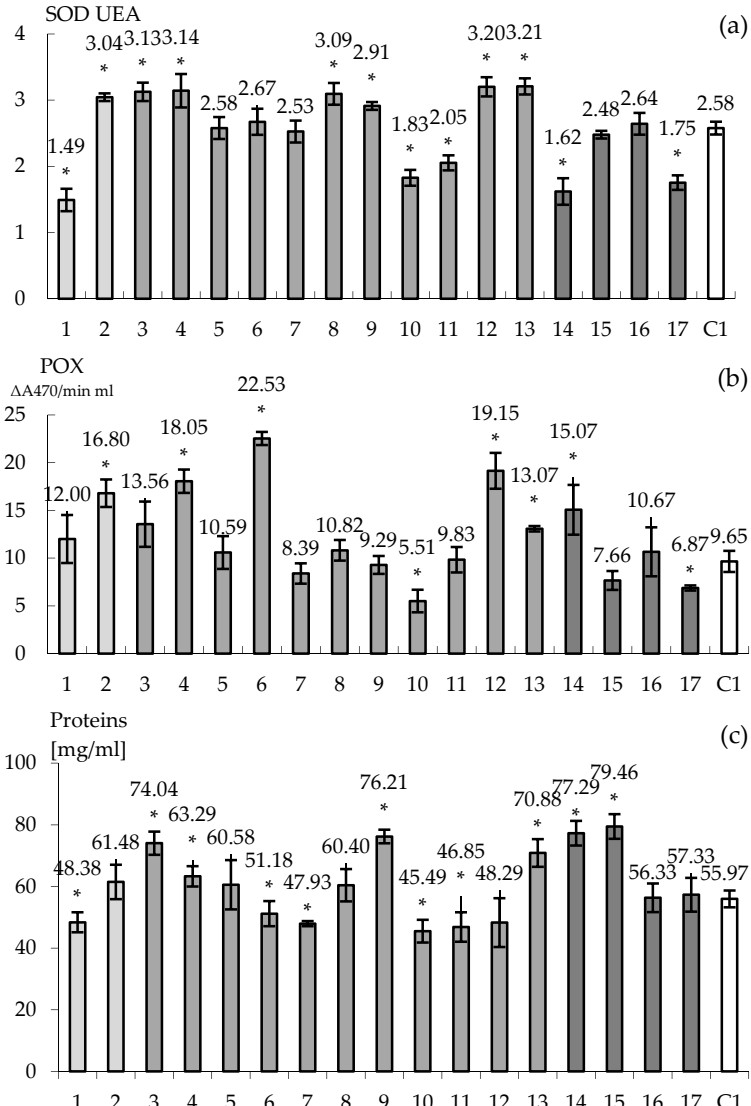

**Figure 6.** The effect of pre-treatment with ABA and/or PEG on activities of SODs (**a**), POXs (**b**) and total protein content (**c**) of the tobacco plantlets (*N. tabacum* L., 'Baladi') after 10 days of acclimatization. Significant differences: $p < 0.05$. The X axe is for evaluating all 17 clones and control C1. * Significant differences: $p < 0.05$.

PEG treatment was studied for 11 clones and almost all of them develop specific answers regarding the internode length. However, there are two clones (i.e., 10 and 11) that express significant decrease of the internode length that are associated with a significant decreased level of SODs, POXs and proteins compared to control. However, there are also some clones that even expressed a decrease in the length of internode, they are expressing

a higher activation of SODs (3, 4, 8, 9, 12 and 13). For POX only clones 4, 12 and 13 expressed a higher activity compared to control. For all these clones the content of protein is higher compared to the control (Figure 6). Activating the oxidative stress by using PEG may further act at the leaf transpiration level [13] but further investigations need to be conducted to prove it. It can be hypostatized that SODs and POXs patterns of expression may further support the need to understanding the acting level at the cellular level of the PEG in such experimental systems.

In the case of ABA & PEG, four clones have been studied. Additionally, all plantlets expressed a decreased internode length. The content of protein is also higher compared to the control and there are differences related to enzymes activities too. Thus, clone 14 has significant lower SOD activity and significant higher POX activity compared to control. Clones 16 and 17 present lower SOD and POX activities also connected with a content of proteins that are comparable with control (Figure 6). However, these differences of SOD and POX activities compared to the control seem to be reflected to be reflected to the internodes height of tobacco shoots. The smaller internode length is the higher POXs activities is. In the case of SOD, this activity may be lower as well as higher compared to the control and further investigation need to be analyzed. It became obvious that ABA and PEG are activating the oxidative stress in plants differently and it becomes obvious that they may act synergically for supporting further morphogenesis.

Considering the results of this experimental model for testing abscisic acid (ABA) and polyethylene-glycol (PEG) effects on caulogenesis, micropropagation and acclimatization of tobacco plantlets it is relevant to emphasize certain concluding remarks. The phenotype of dwarfism is significantly expressed by all clones treated with ABA or/and PEG during all stages of our experiment namely: caulogenesis, micropropagation and acclimatization. Both ABA and/or PEG reduce the weight range of shoot generating hypocotyls but stimulates the production of small shoots as rate of micropropagation. The lack of tobacco shoot rooting for PEG presence into the culture medium and the manifestation of hyperhydration clearly impede the hydric balance in treated tobacco plantlets. This should be considered for future investigations for rooting of more recalcitrant plants for in vitro culture.

If we consider the analysis of our results regarding SOD activity for the first compared to the last stage of experiment it can be considered that for most of tobacco clones SOD activity diminished during time, such as an adaptation mechanism to drought stress. It is not the case for POX, where the level of expression is maintained at high level but also with certain exceptions. As the protein content is not higher it seems that the enzymatic activity is specifically regulated either by lowering or enhancing their activities during acclimatization due to previous treatment with ABA and/or PEG. The oxidative stress study is complex as activators such as ABA and PEG act differently at cellular level. If we only consider hypocotyl morphogenesis, the oxidative stress is higher compared to fully developed plantlets in acclimatization. Under such stress conditions this well differentiated organ started to dedifferentiate and differentiate for producing new shoots. It can be considered that ABA and PEG, under certain controlled conditions may act synergically for supporting morphogenesis as well as for controlling oxidative stress in such a way plant may further survive. However, PEG is not supporting root formation, and this might be due to osmotic potential acting at the interface to plant tissue that is detrimental towards rooting. The same synergic action is seen in acclimatation if we are taking into consideration the pre-treatment with ABA and/or PEG. However, to observe such an effect it is highly important to work with appropriate concentrations of these activators, already proved by previous studies for their morphogenetic effect. Moreover, to couple in perfect controlled environments, morphometry and biochemical analysis is compulsory to prove this synergic activity of both stressors even is too hard to investigate a huge number of samples for a long period of time (i.e., 7 months). In the end we may consider that our experimental results will further support the expression of *drought memory*, a concept that should be further accessed for breeding drought resistance crops as well as in improving the success for acclimatization. The selection of breeding material will remain a challenging

step and such experiments may make easier the effort of breeders for selecting the most appropriate germplasm in terms of drought tolerance [59].

**Author Contributions:** Conceptualization, M.-M.A.; methodology, M.-M.A.; software, C.S.S.; validation, M.-M.A.; formal analysis, C.S.S.; investigation, C.S.S.; resources, M.-M.A.; data curation, M.-M.A.; writing—original draft preparation, M.-M.A.; writing—review and editing, M.-M.A.; visualization, C.S.S.; supervision, C.S.S.; project administration, M.-M.A.; funding acquisition, M.-M.A. All authors have read and agreed to the published version of the manuscript.

**Funding:** This research was funded by the "Lucian Blaga" University of Sibiu and "Hasso Plattner Foundation", internal grant in 2020.

**Institutional Review Board Statement:** Not applicable.

**Informed Consent Statement:** Not applicable.

**Data Availability Statement:** Not applicable.

**Conflicts of Interest:** The authors declare no conflict of interest. The funders had no role in the design of the study; in the collection, analyses, or interpretation of data; in the writing of the manuscript, or in the decision to publish the results.

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
