# Peer review of "Drought Stress Study on Nicotiana tabacum L., “Baladi”, an In Vitro Experimental Model"

_agriculture, doi:10.3390/agriculture11090845_

Round 1
Reviewer 1 Report
The aim of the work was to investiagte the effects of drought responses (through ABA and PEG treatments) on tobacco regeneration in vitro. The authors showed that ABA and PEG increased shoot production, but decreased rooting for PEG, and this was associated with generally increased activity of enzymes associated with response to reactive oxygen species compared to controls, though SOD activities were reduced under some conditions.
The work makes appropriate use of statistics.
I think the manuscript could be improved by 1) some discussion of how ABA and PEG might lead to changes in ROS pathway enzymes; 2) why this is associated with more shoot formation but reduced root formation in vitro; 3) discuss better what the changes in enzymes isoforms might mean in terms of ROS pathway function; and 4) discuss better the reasons for different responses seen in different clones - i.e. are the clones genetically identical or are they somaclonal variants, and why do some respond differently to others?
Author Response
Dear Professor
I do appreciate your effort in providing clues to improve the quality of my paper. In this regard,
1) some discussion of how ABA and PEG might lead to changes in ROS pathway enzymes; I added some discussion for the first stage and the last
2) why this is associated with more shoot formation but reduced root formation in vitro; - I provided some explanations in track changes into the text
3) discuss better what the changes in enzymes isoforms might mean in terms of ROS pathway function; I also provided into the text some more debate in this regards
and
4) discuss better the reasons for different responses seen in different clones - i.e. are the clones genetically identical or are they somaclonal variants, and why do some respond differently to others? I added in the final sentence that we need further analysis to evaluate this.
Reviewer 2 Report
Dear authors,
In general, manuscript is very well written but reader can notice the different way of writing between parts of paper. It is noticeable that two authors written different parts and tried to merge it but they didn´t unify paper at the end. I recommend that the whole paper be checked again and unify way of writing SI units, names of chemicals and abbreviations in text, tables and figures. Special attention should be paid to the excess or lack of whitespace when writing units.
- Row 157 – instead of mEm-2s-1 it should stand mE/m2s (most units are written so)
- 109 - instead of (20µM) it should stand (20 µM) (space)
- 109 - instead of (50g/l) it should state (50 g/l) (space)
- 105; 106; 111; 114 - instead of MS62 it should stand MS 62 (space)
- 121; 125 - instead of 16 / 8h it should be indicated 16 / 8 h; light / dark
- 121; 126 - instead of 53.8µmoles / m2s it should stand 53.8 µmoles/m2s
- Table 1 (6 and 7 column) - instead of ABA (20µM) it should stand ABA (20 µM) (space)
- 137; Fig. 2b; 6b, 183 - instead A 470 nm / min x ml it can be written Δ A/min ml and the author can explain in one sentence that measurements was on 470 nm wavelength in chapter 2
- 141; 144 – instead 4 °C (+2 °C ) it should stand 4°C (+2°C)
- Figure 2: 6b - POX and GPOX abbreviation should be unified
- 294 – units should be stated
- 304 - instead ABA and or PEG it should stand ABA and / or PEG
- 198 - instead 5% (w/v) l PEG 6000 should be written 5% (w/v) PEG6000
- 98; 109 - ABA and/or PEG6000 phrase should be unified
- 94 - culture media MS62 stage 3 perhaps authors mean stage 2
- In Figure 1 on the x-axis numbers are written with a comma instead of a period. 1c – above bar 4-4.9 there is no need for writing 0.00
- Figure 2b – numbers on last bar are unclear (8…); there is no * for indicating significance
- Figure 2b: 2c - axis labels are broken and unreadable
- Figure 4. - is not well described, subtitle should be checked
- SI unit for light intensity is Lux and µmoles/m2s is unit for photon flux density. In row 121 and 126 should be written photon flux density or corrected to Lux (light intensity).
- In several cases, the reference mark in the text do not match the reference mark given at the end of the paper. For example:
- Row 92 - Murashige-Skoog solidified culture medium (MS 1962) without hormones [31] – instead of 31 it should be written 35 because in Row 534 - Murashige, T., & Skoog
- Row 133 - instead of 32 it should stand 31
- Row 134 - instead of 33 it should stand 32
- Row 137 - instead of 34 it should stand 33
- Row 159 - instead of 35 it should stand 34
I recommend that all references should be re-checked and possible errors need to be corrected.
- Not all references are written in the same style (e.g. reference 59). It should be re-checked and possible errors need to be corrected.
- In references where there is Latin name of the species, the name should to be written in italic (e.g. reference 22; 26)
- There should be full list of authors in references and not be marked as three dots (e.g. reference 1; 4; 16; 45; 50).
Suggestions to the authors:
Chapter Materials and Methods were written a little bit messy. Many times authors repeat themselves in different way and lots of crucial information are written in chapter Results and Discussion. It is unclear why did authors use MS62 modified with BAP and NAA in stage 1 and later they didn’t have any treatment with them and didn’t take any explants from that variant. In this chapter, it should be stated clearly how many times and when did authors take plant tissue for biochemical analysis, how many repetitions they had. It is unclear why did they settled 11 clones on PEG and only 1 on Control. It is possible that Control is statistical out layer.
Please try rewriting it succinctly and clearly.
It would be good to state which statistical program authors used for calculation of F test and LSD.
Author Response
Dear professor
I do appreciate your effort in supporting the development of our article to a higher level of clarity. I answered positively to all items in the new draft
